# Death Receptor DR5 as a Proviral Factor for Viral Entry and Replication of Coronavirus PEDV

**DOI:** 10.3390/v14122724

**Published:** 2022-12-06

**Authors:** Xiu-Zhong Zhang, Wen-Jun Tian, Jing Wang, Jing-Ling You, Xiao-Jia Wang

**Affiliations:** Key Laboratory of Animal Epidemiology of the Ministry of Agriculture, College of Veterinary Medicine, China Agricultural University, Beijing 100193, China

**Keywords:** PEDV, death receptor 5, apoptosis, viral entry

## Abstract

Porcine epidemic diarrhea virus (PEDV), a member of *Coronaviridae*, causes high mortality in newborn piglets, and has caused significant economic losses in the pig industry. PEDV infection can induce apoptosis, both caspase-dependent and caspase-independent, but the details of apoptosis remain clarified. This study investigated the effect of death receptor DR5 on PEDV infection and its relationship with PEDV-induced apoptosis. We found that DR5 knockdown reduced viral mRNA and protein levels of PEDV, and the viral titer decreased from 10^4.5^ TCID_50_ to 10^3.4^ TCID_50_ at 12 hpi. Overexpression of DR5 significantly increased the viral titer. Further studies showed that DR5 facilitates viral replication by regulating caspase-8-dependent apoptosis, and the knockdown of DR5 significantly reduced PEDV-induced apoptosis. Interestingly, we detected a biphasic upregulation expression of DR5 in both Vero cells and piglets in response to PEDV infection. We found that DR5 also facilitates viral entry of PEDV, especially, incubation with DR5 antibody can reduce the PEDV binding to Vero cells. Our study improves the understanding of the mechanism by which PEDV induces apoptosis and provides new insights into the biological function of DR5 in PEDV infection.

## 1. Introduction

Porcine epidemic diarrhea virus (PEDV), belonging to the family *Coronaviridae*, is the etiological agent of porcine epidemic diarrhea (PED) [1,2]. The genome of PEDV is approximately 28 Kb in length and encodes pp1a, pp1ab, and ORF3 proteins, as well as four structure proteins: spike protein (S), envelope protein (E), membrane glycoprotein (M), and nucleocapsid protein (N) [3,4]. The pp1a and pp1ab proteins can be cleaved into 16 nonstructural proteins (nsp), named nsp1-16, by viral proteases [5]. PEDV mainly infects the porcine’s small intestinal epithelial cells and leads to pronounced villous atrophy, resulting in vomiting, diarrhea, anorexia, and even death [6,7]. PEDV was first reported in European countries in the 1970s, and then it spread to many countries all over the world [1,4,8,9,10]. Highly pathogenic variant PEDV strains have been reported in China, causing up to 100% mortality in newborn piglets [10]. Since then, highly pathogenic PEDV strains have caused serious problems and huge economic losses in the swine industry [11,12,13].

Apoptosis, also called programmed cell death, is strictly regulated, and is responsible for the elimination of unwanted, injured, or infected cells [14,15]. Apoptosis is a natural process in growing and aging, but it can also be activated by incidental extracellular and intracellular factors or conditions, such as temperature, UV radiation, exposure to alcohol, or bacterial or viral infection [16,17]. Apoptosis is an important innate host defense mechanism and contributes to defense against viral infection [18]. However, many viruses have developed strategies to modulate apoptosis in order to complete their own replication [17,19]. On one hand, viruses can utilize viral proteins or hijack host proteins to inhibit apoptosis, preventing premature cell death, thus allowing the viruses enough time to produce maximal progeny virus [19]. On the other hand, viruses have the ability to induce apoptosis to enhance the release and dissemination of viral progeny to neighboring cells [20]. PEDV infection has been reported to induce apoptosis in both a caspase-independent and a caspase-dependent manner. In Vero cells, PEDV can activate the p53-PUMA signaling pathway, the AIF-associated mitochondrial pathway, or the caspase-8 and -3 associated caspase-dependent pathway [20,21,22]. In Vero cells, the PEDV S protein is a main apoptotic-inducing protein, while the ORF3 protein inhibits apoptosis to promote virus proliferation [19,22]. Apoptosis has also been described in intestinal epithelial cells and intestinal porcine epithelial cell line-J2 cells [23,24]. 

Up to now, the extrinsic/death receptor (DR) pathway and the intrinsic/mitochondrial pathway that are caspase-dependent are the main apoptotic pathways, although caspase-independent apoptosis, such as apoptosis-inducing factor (AIF) and p53-mediated apoptosis has also been identified [14,21,25,26]. Death receptor DR5, also called tumor necrosis factor receptor (TNFR) superfamily member 10b (TNFRSF10B), belonging to the Type I membrane surface receptors, and its ligands TNFSF10/TRAIL belonging to the Type II membrane proteins [27], have also been reported to be implicated in viral infection-related apoptosis [28,29]. Upon TRAIL binding, DR5 recruits the adapter protein Fas-associated death domain (FADD) to form a death-inducing signaling complex (DISC), then associates with pro-cysteinyl aspartic acid protease (caspase)-8, and this leads to the dimerization of caspase-8, resulting in the formation of mature caspase-8, then mature caspase-8 activates caspase-3, which executes apoptosis [27,30]. It has been reported that DR5 takes part in apoptosis induced by several viruses, such as the Hantaan virus, Epstein–Barr virus, and hepatitis B virus [28,29].

Although it is reported that PEDV can induce apoptosis, the details of the apoptotic mechanism remain largely clarified. The relationship of DR5 with PEDV infection was unclear. The expression of DR5-like in piglets under the PEDV challenge was also not mentioned in previous research. The amino acid sequence of DR5-like in *Sus scrofa* shared 43% and 40% homology with DR5 in *Chlorocebus sabaceus* and DR5 in *Homo sapiens*, respectively. In this study, we found that the expression of DR5 protein is significantly increased in PEDV-infected cells and piglets, and a decrease in DR5 greatly reduces the production of progeny PEDV. Further research showed that DR5 is related to caspase-8-dependent apoptosis, which PEDV induced. To our knowledge, we have found the first evidence that DR5 facilitates the early entry of PEDV. In addition, similar to pAPN, a controversial receptor of PEDV, the DR5-like mRNA level was significantly increased in piglets at the early stage of PEDV infection. These findings expand our understanding of PEDV-induced apoptosis and the biological function of DR5 in PEDV infection.

## 2. Materials and Methods

### 2.1. Cells, Viruses, Reagents and Antibodies

Vero E6 cells were obtained from American Type Culture Collection (ATCC) and cultured in Dulbecco’s modified Eagle’s medium (DMEM) (Gibco Invitrogen, Carlsbad, CA, USA) supplemented with 5% or 10% fetal bovine serum (FBS) and antibiotics (100 U/mL penicillin and 100 mg/mL streptomycin) at 37 °C in a humidified 5% CO_2_ incubator (all reagents obtained from Invitrogen, Carlsbad, CA, USA). Herpes simplex virus-1 (HSV-1) strain F, vesicular stomatitis virus (VSV) strain Indiana, and porcine epidemic diarrhea virus (PEDV) strain CV777 were reproduced in Vero cells for 48 h. Rabbit monoclonal anti-DR5 were from Abcam (Cambridge, MA, USA). Human anti-Rabbit antibody-cleaved caspase-8 (p43), human anti-Rabbit antibody-cleaved caspase-3, and mouse monoclonal anti-FLAG was from Cell Signaling Technology (Danvers, MA, USA). The mouse monoclonal anti-GAPDH, FITC-conjugated goat anti-Mouse antibody, TRITC-conjugated goat anti-mouse antibody was from Proteintech (Rosemount, IL, USA). Mouse monoclonal anti-PEDV-N was from Alpha Diagnostic International (San Antonio, TX, USA). The pharmacological inhibitors Z-LE(OMe)TD(OMe)-FMK and Rh2 were from MedChemExpress (Monmouth Junction, NJ, USA). Annexin V- FITC/PI, Dead Cell Apoptosis kit was from Solarbio (Beijing, China). 

### 2.2. Virus Titration

The viral titration of PEDV was performed in 96-well plates using 50% tissue culture infective dose (TCID_50_) with the method of Karber, and assays according to the procedure described previously [31]. Briefly, Vero cells were seeded into the 96-well plates at a density of 10^5^ cells/well. Then the medium was removed, ten-fold serial dilutions were prepared for each sample, and 100 μL/well of each dilution was added to the cells in quadruplicate. The cytopathic effect was examined every 24 h for 4 days post-inoculation.

### 2.3. RNA Interference (RNAi) and Cell Viability Assay

Small interfering RNAs (siRNAs) targeting DR5 (sense: 5′-CCCUGAACAGGAAAUGGAATT-3′, antisense: 5′-UUCCAUUUCCUGUUCAGGGTT-3′) and negative control (NC) siRNA were synthesized by Genepharma (Shanghai, China). Cells grown to 60–70% confluence were transfected with siRNAs (final concentration of 100 nM) using Lipofectamine RNAi MAX (Invitrogen, Carlsbad, CA, USA). At 48 h post-transfection, cells were used for subsequent experiments.

### 2.4. Recombinant Plasmid and Transfection

The vector pcDNA3.1 was purchased from Clontech. FLAG-tagged DR5 was synthesized by Shanghai General Biotech and cloned into pcDNA3.1. Cells grown to 70–90% confluence were transfected with recombinant plasmid or empty vector using Lipofectamine™ LTX (Invitrogen, Carlsbad, CA, USA). At 24 h post-transfection, cells were used for subsequent experiments.

### 2.5. Total RNA Extraction and Quantitative Real-Time PCR

According to the manufacturer’s protocol, total RNA was extracted from cells cultured using TRIzol reagent (Invitrogen, Carlsbad, CA, USA), and reverse transcription was conducted using One-Step gDNA Removal and cDNA Synthesis SuperMix (TransGen Biotech, Beijing, China). From three independent experiments, transcription levels for different genes were calculated using a Bio-Rad PCR instrument (Hercules, CA, USA) and the SYBR green supermix (Toyobo, Osaka, Japan). The following primers were used: PEDV-M forward: GATACTTTGGCCTCTTGTGT and PEDV-M reverse: CACAACCGAATG CTATTGACG; DR5 forward: GTTGTGGTTGTGGCTGTGATTGTT and DR5 reverse: TGTCTGCTCTGCTGGCTCCT; GAPDH forward: GTCTTCACTACCAT GGAGAAGG and reverse: TCATGGATGACCTTGGCCAG; DR5-like forward: GCACAGCAACTGCGGACACA and DR5-like reverse: GCACAGCAACTGCGG ACACA. The RNA levels of viral genes were normalized by GDAPH mRNA, and relative quantities (RQ) of mRNA accumulation were evaluated using the 2^−ΔΔCt^ method.

### 2.6. Western Blot Analysis

Cells were washed in ice-cold PBS, then lysed and harvested in Radio-Immunoprecipitation Assay buffer (RIPA buffer) in the presence of protease inhibitor cocktail. Protein concentrations were measured using a BCA Kit (Beyotime, Shanghai, China), and separated by SDS-PAGE on 12.5% gels (Beyotime, Shanghai, China), then electroblotted onto polyvinylidene fluoride (PVDF) membrane. The membranes were incubated with the primary antibody overnight, and the blots were incubated with horseradish peroxidase (HRP)-conjugated secondary antibody for 45 min at room temperature. Finally, the blots were exposed using an ECL detection system (Vazyme, Nanjing, China), and Western blotting bands were quantified according to intensity using ImageJ software.

### 2.7. Immunofluorescence Assay (IFA)

Vero cells transfected with siDR5 or siNC were grown on microscope coverslips placed in 12-well plates, and were infected with PEDV at an MOI of 0.1 for 1 h at 37 °C. The virus inoculum was subsequently discarded and washed by PBS three times, then replaced by a normal growth medium. At 18 hpi, the cells were fixed with 4% paraformaldehyde for 15 min at RT, permeabilized with 0.3% Triton X-100 in PBS at RT for 10 min, and the non-specific binding was blocked with 5% bovine serum albumin (BSA). Then the cells were incubated with mouse anti-PEDV N monoclonal antibody (1:100) overnight at 4 °C. After being washed three times with PBS, the cells were incubated for 45 min at RT with a FITC-conjugated goat anti-mouse secondary antibody (1:100). The cells were again washed three times with PBS, incubated with rabbit anti-DR5(1:100) overnight at 4 °C, washed three times, incubated for 45 min at RT with a TRITC-conjugated goat anti-mouse secondary antibody (1:100). Finally, the cells were counterstained with 4′,6-diamidino-2-phenylindole (DAPI) (Beyotime, Shanghai, China), and the images were viewed under a Zeiss LSM900 confocal microscope (Fluoview 200×).

### 2.8. Annexin V and PI Staining Assay

To examine the effect of DR5 on apoptosis, cells were grown in 6-well plates, transfected with siRNA targeting DR5 for 24 h, and then infected with viruses at MOI of 0.1 or mock-infected. As a positive control, cells were treated with Ginsenoside Rh2 (inducer of apoptosis by activating caspase-8 and caspase-9) for 1 h at a concentration of 100 μM. At 24 hpi, phosphatidylserine exposure was determined by measuring Annexin V binding using an Annexin V- FITC/PI Dead Cell Apoptosis kit (Invitrogen, Carlsbad, CA, USA), according to the manufacturer’s protocol. Acquisition of the apoptotic cells was quantified by a BD FACSCanto II flow cytometer (BD Biosciences, Franklin Lakes, NJ, USA), and the data were analyzed using the FlowJo Version 10.8.1 software (Ashland, Covington, KY, USA). With the kit, cells only positive for Annexin V were considered early apoptotic, cells only positive for PI were classified as necrotic, and cells with double labeling were categorized as late apoptotic.

### 2.9. PEDV Entry Assay

To investigate the effect of DR5 on the entry of PEDV, cells were grown in 6-well plates, transfected with siRNA targeting DR5 for 36 h, then infected with viruses at MOI of 0.1, 0.5, or 1 at 37 °C for 2 h to allow viral entry, or at 4 °C for 1 h to allow viral binding. E64D (0.1 μM), a cysteine protease inhibitor, was also used; it was pre-added to cells before 1 h viral infection. Cells were harvested for total RNA extraction and Quantitative Real-Time PCR. 

### 2.10. In Vivo Experiment 

Newborn SPF piglets that had not drunk colostrum were obtained and fed special milk to 3 days old. Then piglets were challenged with 10^5^ TCID_50_ PEDV or DMEM in 3 mL by adding it to the milk. Piglets were humanely slaughtered at the pointed time, and the jejunum was obtained. The protein and mRNA levels of DR5-like were detected by Western blot and qRT-PCR. There were three piglets in each group.

### 2.11. Statistical Analysis

All assays were each repeated 3 times, and all results were presented as the mean ± SD. Statistical analyses were performed using Prism Version 7.0 (GraphPad Software, La Jolla, CA, USA). Significance was determined by a one-way analysis of variance (ANOVA) and a two-way ANOVA with Dunnett’s multiple-comparison test. Partial correlation analyses were evaluated using an unpaired Student’s *t*-test. For all analysis, *p* values of <0.05 were considered statistically significant (* *p* < 0.05; ** *p* < 0.01; *** *p* < 0.001; ns *p* > 0.05).

## 3. Results

### 3.1. DR5 Facilitated Viral Infection of PEDV

Considering that DR5 affects the viral production of several viruses [28,29], we first determined the effect of DR5 on the viral replication of PEDV. Vero cells transfected with siDR5 or siNC were infected with PEDV at different times. Western blot analysis showed that the knockdown of DR5 significantly decreases the level of PEDV-N protein, and the relative gray level ratio was reduced to 0.40, 0.50, and 0.79 at 12, 24, and 36 hpi, respectively (Figure 1A). The decrease in DR5 corresponds to a 10-fold decrease in viral titer in TCID_50_ at 12 hpi (Figure 1B). The level of PEDV mRNA was also reduced (Figure 1C,D). The results of IFA revealed that the number of viral particles in siDR5 groups was far lower than that in siNC groups, and DR5 levels were significantly promoted upon PEDV infection (Figure 1E). Finally, transient expression of FLAG-tagged DR5 increased the PEDV-N level (Figure 1F) and viral titer (Figure 1G). The results demonstrate that DR5 enhances the viral production of PEDV in Vero cells.

### 3.2. PEDV Infection Increased the Levels of DR5 mRNA and Protein

We noticed that the expression of DR5 was upregulated upon PEDV infection (Figure 1E), so we characterized the levels of DR5 mRNA and protein in response to viral infection. We detected a biphasic upregulation expression of DR5: DR5 protein (Figure 2A) and mRNA (Figure 2B) were elevated at 1 hpi, decreased to mock-infected levels between 2 hpi and 4 hpi, and then rapidly increased after 8 hpi (Figure 2A,B). We also found that the expression level of DR5 is increased in a virus dose-dependent manner (Figure 2C). In comparison, the protein levels of DR5 remain unchanged in the early and later stages of HSV-1 (Figure 2D) or VSV infection (Figure 2E). The results indicate that PEDV infection specifically enhances the expression level of DR5 protein.

### 3.3. DR5 Activated Caspase-8-Dependent Apoptosis Induced by PEDV Infection

Previous studies showed that caspase-8 was activated upon PEDV infection, and DR5 is known as an apoptosis regulator for activating caspase-8 [27]. Therefore, we hypothesized that DR5 enhances PEDV production by activating caspase-8-dependent apoptosis. To prove this hypothesis, Vero cells transfected with siDR5 or siNC were infected with PEDV, and cultures were obtained at a pointed time. Cells treated for 1 h with Ginsenoside Rh2, an inducer of apoptosis by activating caspase-8 and caspase-9 [32], were used as a positive control. Assessment of cell apoptosis showed that DR5 knockdown reduced the number of early apoptotic cells from 13.1 to 2.87 percent at 24 hpi (Figure 3A). Cleaved caspase-8 and cleaved caspase-3 were detected upon PEDV infection (Figure 3B,C), which is consistent with previous research [22,33]. Furthermore, DR5 knockdown inhibited the activity of caspase-8 and caspase-3. These results indicate that DR5 might regulate apoptosis induced by PEDV infection by activation of caspase-8.

In addition, Vero cells were infected with PEDV at MOI of 0.1 at the presence of Z-LE(OMe)TD(OMe)-FMK (0.1 μM), a selective inhibitor of caspase-8 [34]. We found that Z-LE(OMe)TD(OMe)-FMK reduces the protein level of PEDV-N, which showed that caspase-8-dependent apoptosis is important to the replication of PEDV (Figure 3C). Interestingly, the effects of DR5 knockdown and Z-LE(OMe)TD(OMe)-FMK treatment on the reduction in the PEDV infection were superimposed, which revealed that DR5 might have an additional function (Figure 3C). Our results suggest that DR5 facilitates PEDV replication by contributing to the activation of the death receptor pathway. 

### 3.4. DR5 Promoted the Early Entry of PEDV

Considering that the level of DR5 protein, a membrane protein, was increased at 1 h post-infection (Figure 2A,B), we studied whether DR5 influences the viral entry of PEDV. Vero cells transfected with siDR5 or siNC were infected with PEDV at 37 °C for 2 h to allow viral entry. We found that the knockdown of DR5 significantly reduced PEDV entry (Figure 4A). In addition, siRNA efficiency was assessed as shown in Figure 4B. To determine the stage of viral entry in which DR5 plays a role, cells were treated with E64D (0.1 μM, a cysteine protease inhibitor) 1h before viral infection. The results showed that E64D reduced PEDV entry, and importantly, it showed further inhibition after DR5 knockdown (Figure 4C,D). Our findings suggest that DR5 enhances early entry of PEDV.

Furthermore, Vero cells transfected with siDR5 or siNC were infected with PEDV at 4 °C for 1 hour to allow viral binding. We found that DR5 knockdown impaired viral binding (Figure 4E,F). This coincides with the finding that the binding of PEDV is inhibited by the DR5 antibody (Figure 4G). The binding of HSV-1, on the other hand, was not affected when cells were pre-treated with DR5 antibody (Figure 4H). The results suggest that DR5 facilitates virus attachment of PEDV.

### 3.5. The Expression of DR5-like Gene in Piglets upon PEDV Infection

The expression of DR5-like in piglets under the PEDV challenge was unclear. Therefore, we detected the protein and mRNA levels of DR5-like in PEDV-infected piglets. Regardless of the controversy of pAPN as the functional receptor of PEDV, in this study, we used it as a control. Three-day-old piglets were challenged with PEDV at different times, and then mercy killing was carried out. We found that the protein level of DR5-like was significantly increased at 3 hpi and slightly upregulated at 12 hpi in the jejunum of piglets (Figure 5A), which conforms to the DR5-like mRNA level (Figure 5B). The pAPN mRNA level also significantly increased at 3 hpi and then decreased to a normal level. Interestingly, the protein and mRNA levels of DR5-like were slightly upregulated at a later period of the PEDV challenge (Figure 5), which is different from the DR5 level in Vero cells (Figure 2). Further study will focus on the effect of DR5-like on viral entry in piglets upon PEDV infection, and related mechanisms.

## 4. Discussion

PEDV has been a major threat and caused great economic losses to the global swine industry since highly pathogenic variant strains were reported in 2010 in China. Half a century after its first identification, it is unfortunate that PEDV still has a high prevalence worldwide. From 2017 to 2021, for example, the percentage of positive farms for PEDV was about 52.15% of samples in China [35]. In Mexico, about 67.6% of diarrhea samples from 10 farms were positive [36]. In Spain, 38.7% of farms with diarrhea outbreaks were positive for PEDV between 2017 and 2019 [37]. There currently are no effective vaccines or specific drugs available to treat PEDV [38], and there is, therefore, an urgent need for basic research on the etiology and pathogenesis. In this study, we found that the death receptor DR5 is involved in caspase-8-dependent apoptosis induced by PEDV infection, enhancing viral production (Figure 1 and Figure 3). Our findings enrich the understanding of the apoptosis induced by PEDV infection through the death receptor pathway. We also found that DR5 facilitates PEDV entry into host cells (Figure 4), which provides new insights into the biological function of DR5 in viral PEDV infection. 

In our study, we found that DR5 enhanced the production of PEDV (Figure 1), which was not a unique case of the functioning of DR5 in viral replication. X protein (HBx) encoded by hepatitis B virus (HBV) promoted the association of DR5 with LC3B, resulting in the downregulation of DR5 by the lysosoma, leading to inhibition of TRAIL-mediated induction of apoptosis, and facilitating viral production [39]. The Hantaan virus (HTNV) inhibits extrinsic apoptosis by the degradation of DR5 via 26S proteasome-dependent pathway and hampers DR5 transport to the cell surface [28]. HCV can induce DR4/DR5-dependent activation of caspase-8, causing elevation of apoptotic signaling in infected cells, and exhibits the TRAIL effect in HCV-induced apoptotic signaling [40]. The current study revealed that PEDV induces extrinsic apoptosis by upregulation of DR5, resulting in the activation of caspase-8 in PEDV-infected Vero cells (Figure 2 and Figure 3). This process facilitates PEDV replication.

Apoptosis is an important host innate defense mechanism; nevertheless, there is a bidirectional relationship between cell apoptosis and viral infection. In response to viral infection, host cells eliminate infected cells by inducing cell apoptosis to prevent viral replication. Viruses have acquired resistance to apoptosis, prolonging the time available for viral replication. Viruses also induce apoptosis, to facilitate release and dissemination to neighboring cells. Research has suggested that several coronaviruses can induce apoptosis to enhance viral replication. TGEV, for example, induces both caspase-dependent and caspase-independent apoptosis by activating the p38 MAPK signaling pathway [33,41,42]. PDCoV infection may promote Bax translocation and relocate the mitochondrial cytochrome c into the cytoplasm, which activates caspase-9/3 in ST cells [43]. SARS-CoV infection can downregulate Bcl-2 and upregulate Bax, inducing apoptosis via the mitochondrial signaling pathway [44]. PEDV infection has previously been known to induce both caspase-dependent and caspase-independent apoptosis, but the factor of activation of caspase-8 was previously unknown. Our study showed that PEDV infection can activate caspase-8 and caspase-3 (Figure 3), supporting previous findings [2,20,22]. DR5 protein was significantly upregulated upon PEDV infection in Vero cells, causing caspase-8 activation, and resulting in cell apoptosis. 

Virus-specific receptors mediate the entrance of viruses into host cells. The receptors of most coronaviruses have been identified; HCoV-229E utilizes aminopeptidase N (APN), SARS-CoV and SARS-CoV use angiotensin-converting enzyme 2 (ACE2), MHV enters through CEACAM1, and MERS-CoV employs dipeptidyl-peptidase 4 (DPP4) [45]. Numerous studies have focused on identifying the PEDV cellular receptor, but the results have been inconclusive. Earlier research suggested that pAPN is the functional host receptor of PEDV, based evidence that anti-pAPN antibody disrupted the interaction between PEDV and pAPN in swine testis (ST) and intestinal cells, overexpression of exogenous pAPN increased PEDV infection in ST cells, and the PEDV S1 subunit binds effectively to pAPN by dot blot hybridization [46,47]. Later studies disproved pAPN as the PEDV receptor, because pAPN has no ability to bind PEDV spikes, and pAPN overexpressed or genetically ablated MDCK, Huh7, ST, and HeLa cells, and pAPN rendered Vero cells or IPEC-J2 cells unsusceptible to PEDV infection [48,49]. In this study, we found that the mRNA level of pAPN was significantly increased in piglets at an early stage of PEDV infection (Figure 5), which matches part of the characteristics of the receptor. Studies have identified several other factors that promote PEDV infection, including sialic acid, heparan sulfate (HS), DC-SIGN, occludin, and integrin αvβ3 [50]. Here, we found that DR5 facilitates PEDV binding to host cells (Figure 4). DR5 in Vero cells and in piglets was specifically upregulated in an early stage of viral infection (Figure 2 and Figure 5), and the DR5 antibody prevents PEDV entry (Figure 4). This previously uncharacterized protein of DR5 is involved in the efficient viral entry of PEDV. Death receptor CD140a has been shown to be associated with viral entry of human immunodeficiency virus (HIV-1); TNFα, as CD140a’ ligand, is able to selectively regulate the entry of HIV-1 into macrophages [51]. Whether death receptors cooperate with host receptors to facilitate virus invasion has been hypothesized in SARS-CoV-2 research and is an interesting research direction.

## 5. Conclusions

Our study demonstrates that DR5 facilitates PEDV production via the regulation of caspase-8-dependent apoptosis and viral entry. Further research is needed on the regulatory function of DR5 in the viral replication cycle and on, for instance, the relationship between DR5 and the viral proteins of PEDV that are involved in viral entry and PEDV-induced apoptosis. Altogether, our findings provide a new target for the study of antiviral agents and broaden the view of the biological function of DR5 in PEDV entry and infection.

## Figures and Tables

**Figure 1 viruses-14-02724-f001:**
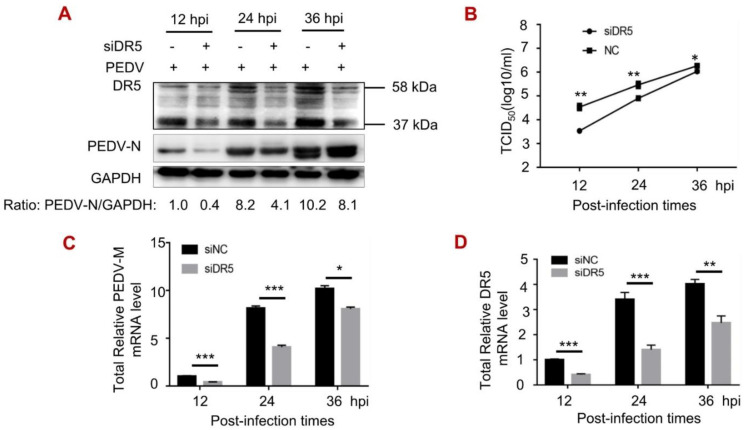
DR5 enhances viral replication of PEDV. (**A**–**E**), Vero cells were transfected with siRNA targeting DR5 (siDR5) or negative control (siNC), and then infected with PEDV at an MOI of 0.1, and cells were harvested at pointed time. The protein level was analyzed by Western blot with the indicated antibodies (**A**). Viral titration was performed using TCID_50_ with the method of Karber (**B**). The mRNA level was subjected to RT-qPCR analysis for PEDV-N (**C**) and DR5 (**D**). For immunostaining (**E**), the infected cells were fixed at pointed time and incubated with mouse anti-PEDV N monoclonal antibody followed by FITC-conjugated goat anti-mouse secondary antibody (green). Then cells were incubated with rabbit anti-DR5 followed by TRITC-conjugated goat anti-mouse secondary antibody (1:100). Finally, the cells were counterstained with DAPI (blue), and the images were viewed under a Zeiss LSM900 confocal microscope (Fluoview 200×). (**F**,**G**), Vero cells were transfected with empty vector pFlag or recombinant plasmid pFlag-DR5. At 24 h post-transfection, cells were infected with PEDV at an MOI of 0.1, and cells were harvested at pointed time. The protein level was analyzed by Western blot with the indicated antibodies (**F**). Viral titration was performed using TCID_50_ with the method of Karber (**G**). *, *p* < 0.05; **, *p* < 0.01; ***, *p* < 0.001.

**Figure 2 viruses-14-02724-f002:**
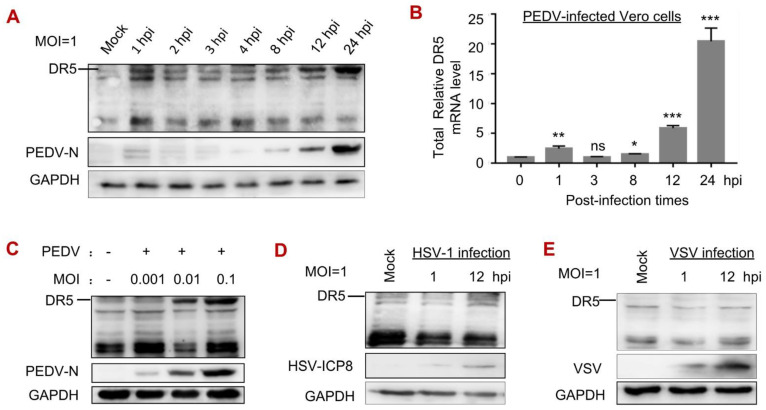
DR5 expression is upregulated by viral infection of PEDV but not HSV-1 or VSV. (**A**,**B**), Vero cells were mock infected or infected with PEDV at MOI of 1, cells were harvested at pointed time. The protein level was analyzed by Western blot with the indicated antibodies (**A**). The mRNA level was subjected to RT-qPCR analysis for DR5 (**B**). (**C**) Vero cells were mock infected or infected with PEDV at MOI of 0.001, 0.01, and 0.1. At 24 hpi, cells were harvested and the protein level was analyzed by Western blot with the indicated antibodies. (**D**,**E**) Vero cells were mock infected or infected with HSV-1 (**D**) or VSV (**E**) at MOI of 1. At 24 hpi, cells were harvested and the protein level was analyzed by Western blot with the indicated antibodies. *, *p* < 0.05; **, *p* < 0.01; ***, *p* < 0.001; ns *p* > 0.05.

**Figure 3 viruses-14-02724-f003:**
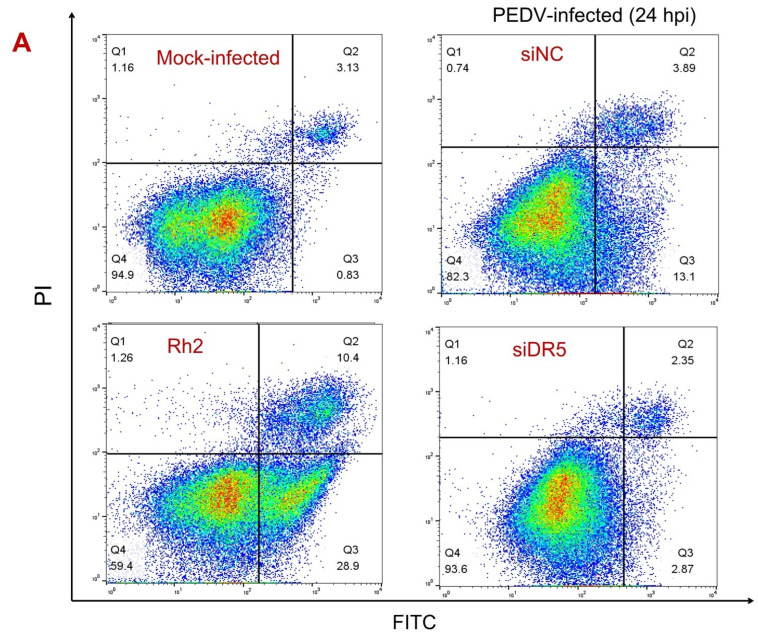
DR5 is associated with caspase-8-dependent apoptosis in PEDV-infected Vero cells. (**A**), Cell apoptosis was analyzed by flow cytometry with dual AnnexinV-PI cell labeling. Vero cells were transfected with siDR5 or siNC for 24 h, and then mock infected or infected with PEDV at MOI of 0.1. At 24 hpi, cells were collected and dual-labeled with AnnexinV and PI and analyzed by FACS. Lower left quadrants represent intact cells; lower right quadrants represent early apoptotic cells; upper right quadrants indicate late apoptotic cells; upper left quadrants indicate necrotic cells. Figure representative of three independent experiments. Graph on the right represents the percentage of apoptotic cells. As positive control, cells were pretreated with Ginsenoside Rh2 (inducer of apoptosis) at a concentration of 100 μM for 1 h. (**B**,**C**), Vero cells were transfected with siDR5 or siNC for 24 h, and then mock infected or infected with PEDV at MOI of 0.1, in the presence of Ginsenoside Rh2 (inducer of apoptosis) (**B**), Z-LE(OMe)TD(OMe)-FMK (0.1 μM), or DMSO (0.1%) (**C**). At pointed time, cells were harvested and the protein level was analyzed by Western blot with the indicated antibodies. Note: cells were pretreated with Ginsenoside Rh2 for 1 h.

**Figure 4 viruses-14-02724-f004:**
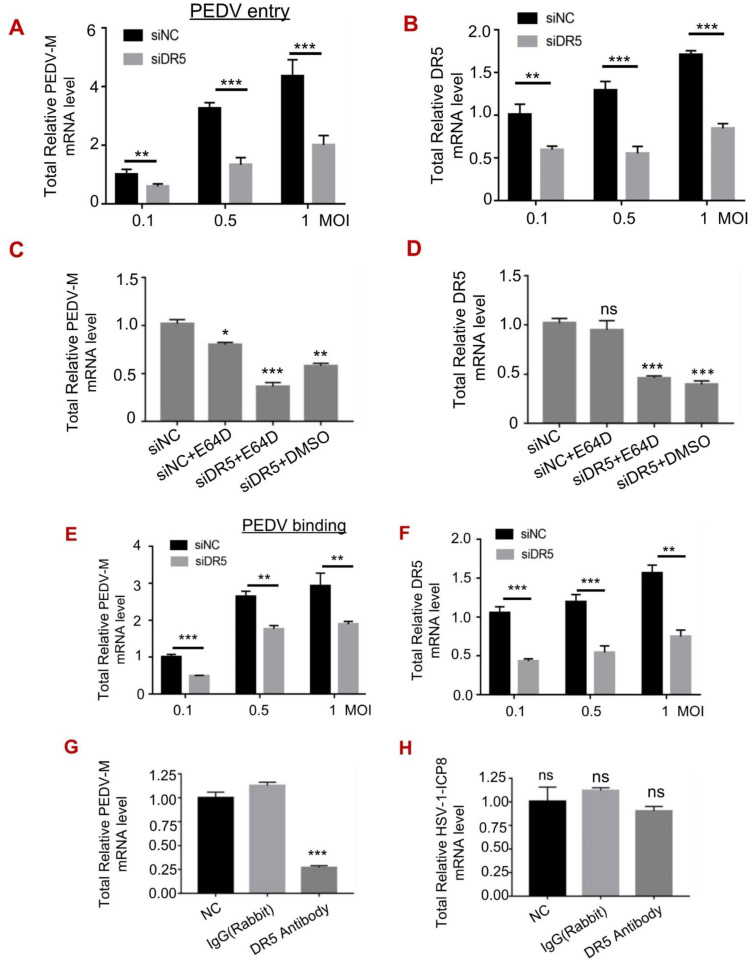
DR5 facilitates viral entry of PEDV into Vero cells. (**A**,**B**), Vero cells were transfected with siDR5 or siNC for 36 h, and infected with PEDV at an MOI of 0.1, 0.5, or 1 at 37 °C for 2 h. After washing with PBS, the infected cells were harvested. The mRNA levels of PEDV m (**A**) and DR5 (**B**) were subjected to RT-qPCR analysis. C-D, Vero cells were transfected with siDR5 or siNC for 36 h, and were pre-treated with E64D (0.1 μM) for 1 h before viral infection. Cells were then infected with 1 MOI PEDV at 37 °C for 2 h and the mRNA levels of PEDV m (**C**) and DR5 (**D**) were subjected to RT-qPCR analysis. (**E**,**F**), Vero cells were transfected with siDR5 or siNC for 36 h, and then infected with PEDV at an MOI of 0.1, 0.5, or 1 at 4 °C for 1 h. The mRNA levels of PEDV m (**E**) and DR5 (°C) were subjected to RT-qPCR analysis. G-H, Vero cells were pre-treated with rabbit antibody against DR5 (1:100) or rabbit IgG (1:100) for 2 h. After washing with PBS, cells were infected with 1 MOI PEDV or HSV-1 at 4 °C for 1 h. The mRNA levels of PEDV m (**G**) and HSV-1-ICP8 (**H**) were subjected to RT-qPCR analysis. *, *p* < 0.05; **, *p* < 0.01; ***, *p* < 0.001; ns *p* > 0.05.

**Figure 5 viruses-14-02724-f005:**
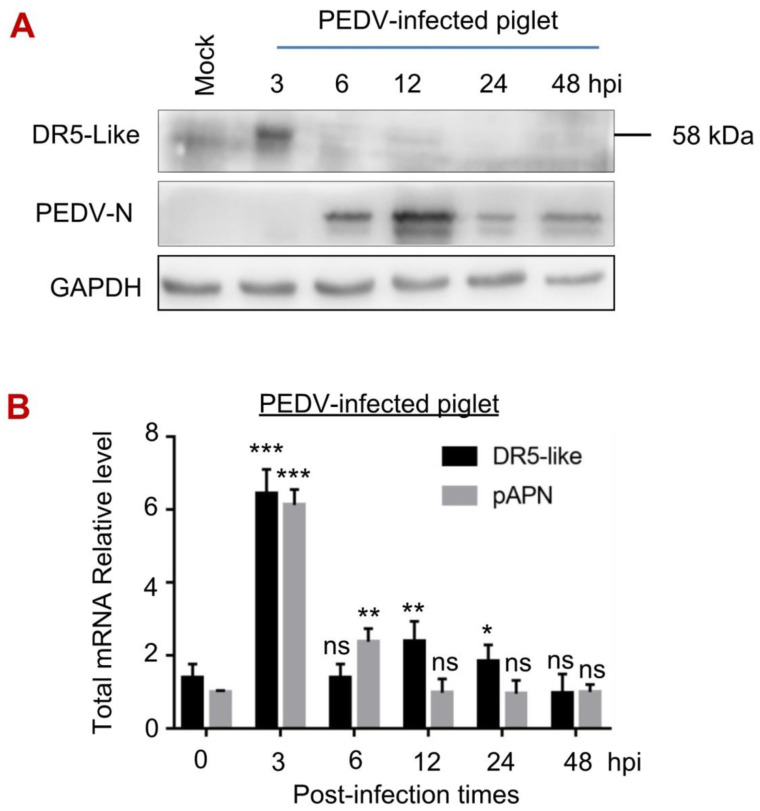
Levels of DR5-like mRNA and proteins in jejunum of piglets during PEDV infection. (**A**,**B**), Piglets were challenged with 10^5^ TCID_50_ PEDV or DMEM in 3 mL for different times, and the jejunum was obtained. The protein level was analyzed by Western blot with the indicated antibodies and mRNA level of DR5-like was analyzed by RT-qPCR. *, *p* < 0.05; **, *p* < 0.01; ***, *p* < 0.001; ns *p* > 0.05.

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
