# Peer review of "Death Receptor DR5 as a Proviral Factor for Viral Entry and Replication of Coronavirus PEDV"

_viruses, 2022, doi:10.3390/v14122724_

Round 1

Reviewer 1 Report

In the manuscript “Death receptor DR5 as a proviral factor for viral entry and replication of coronavirus PEDV”, the authors investigated the role of death receptor 5 (DR5) in the viral entry and replication of porcine epidemic diarrhea virus (PEDV. The authors performed silencing of DR5 and assessed the infection of PEDV in Vero cells by measuring protein levels of PEDV-N and mRNA levels of PEDV-M, as well as viral titers, which showed a decrease in viral replication. The overexpression of DR5 resulted in increased PEDV infection. They showed upregulation of DR5 upon PEDV infection but no other viruses such as HSV-1 and VSV. They also found that DR5 might induce apoptosis via caspase-8. Next, the authors investigate the role of DR5 in viral entry and binding, the results indicate the participation of DR5 in both. Lastly, authors investigated the expression of DR5-like in piglets and found it upregulated upon 3 h post-PEDV infection decreasing thereafter. With these results, the authors demonstrated a direct relationship between DR5 and PEDV. However, some issues need clarification.

1.       In the abstract section, authors indicate they “detected a biphasic up-regulation expression of DR5 in both Vero cells and piglets in response to PEDV infection” which was not mentioned either in the results or discussion.

2.       In the introduction, authors should indicate that DR5 has not been described in pigs, since there is no hint of that up to results from Figure 5, where they indicate the finding of a DR5-like protein and mention homology with DR5 from Chlorocebus. The authors should indicate the homology with other species, such as humans

3.       Authors should also indicate the role of pAPN in PEDV infection. It was mentioned in the Discussion, but they should introduce it before in the manuscript In order to make sense for its use in later experiments (Figure 5)

4.       In Materials and methods, authors should clarify the host and reactivity of antibodies used for western blot, IFA, and Flow cytometry.

5.       In results, Figure 1 A showed 2 bands at 58  and 37 KDa, clearly indicated in the western blot image, but the authors did not discuss this finding; what is the implication? are they isoforms?

6.       Related to Fig 5A, is DR5-like the same molecular size? Could you please indicate in the western blot

Some typographical errors detected:

Page 2 Line 49, One hand … should be “On one hand, …”

P3 L119, pCDNA3.1 should be “pcDNA3.1” as stated everywhere

P4 L155, the word with is repeated with with

P4 L174, cells only positive for PI were classified as normal, should be “cells only positive for PI were classified as necrotic”. 

Author Response

Viruses

Manuscript ID: viruses-2056518

Title: Death receptor DR5 as a proviral factor for viral entry and replication of coronavirus PEDV

Response to Reviewer 1 Comments

Reviewer 1 comments and Suggestions for Authors

In the manuscript “Death receptor DR5 as a proviral factor for viral entry and replication of coronavirus PEDV”, the authors investigated the role of death receptor 5 (DR5) in the viral entry and replication of porcine epidemic diarrhea virus (PEDV. The authors performed silencing of DR5 and assessed the infection of PEDV in Vero cells by measuring protein levels of PEDV-N and mRNA levels of PEDV-M, as well as viral titers, which showed a decrease in viral replication. The overexpression of DR5 resulted in increased PEDV infection. They showed upregulation of DR5 upon PEDV infection but no other viruses such as HSV-1 and VSV. They also found that DR5 might induce apoptosis via caspase-8. Next, the authors investigate the role of DR5 in viral entry and binding, the results indicate the participation of DR5 in both. Lastly, authors investigated the expression of DR5-like in piglets and found it upregulated upon 3 h post-PEDV infection decreasing thereafter. With these results, the authors demonstrated a direct relationship between DR5 and PEDV. However, some issues need clarification.

Response: We are submitting a revised manuscript to you to consider for publication in Viruses. In the updated manuscript, we have fully addressed reviewer comments, point-by-point, with an updated figure, and additional information on DR5-like and pAPN in the Introduction section, which are required for further clarification in the manuscript.

Minor points:

Point 1:  In the abstract section, authors indicate they “detected a biphasic up-regulation expression of DR5 in both Vero cells and piglets in response to PEDV infection” which was not mentioned either in the results or discussion.

Response: We appreciate the reviewer’s comment. The “biphasic up-regulation expression of DR5” means that the expression of DR5 was up-regulation at 1 hpi and after 8 hpi, and it might relate with the function of DR5 in viral entry and viral induced apoptosis, respectively.  These parts were demonstrated in the sections Result of 3.2 and Discussion of the last two paragraphs.

Point 2: In the introduction, authors should indicate that DR5 has not been described in pigs, since there is no hint of that up to results from Figure 5, where they indicate the finding of a DR5-like protein and mention homology with DR5 from Chlorocebus. The authors should indicate the homology with other species, such as humans

Response: In the Introduction, we added the information of DR5-like protein in swine, and the amino acid sequence of DR5-like in Sus scrofa shared 43% and 40% homology with DR5 in Chlorocebus sabaceus and DR5 in Homo sapiens, respectively .

Point 3: Authors should also indicate the role of pAPN in PEDV infection. It was mentioned in the Discussion, but they should introduce it before in the manuscript In order to make sense for its use in later experiments (Figure 5)

Response: We added it in the sections Introduction, Results, and Discussion.

Point 4: In Materials and methods, authors should clarify the host and reactivity of antibodies used for western blot, IFA, and Flow cytometry.

Response: Thanks, we added the details in the section Materials and methods.

Point 5: In results, Figure 1 A showed 2 bands at 58 and 37 KDa, clearly indicated in the western blot image, but the authors did not discuss this finding; what is the implication? are they isoforms?

Response: The DR5 has 2 bands in the western blot; the 37 kDa band is an alternatively spliced variant of DR5. This is a common result, so there was no special instruction in most articles. We added this information in our LEGENDS of Figure 1.

Point 6: Related to Fig 5A, is DR5-like the same molecular size? Could you please indicate in the western blot

Response: We have indicated the molecular size of DR5-like in pigletsï¼›and the size is 58 kDa. No band with 37 kDa was observed.

Point 7: Page 2 Line 49, One hand … should be “On one hand, …”

Response: Thanks, we have corrected it.

Point 8: P3 L119, pCDNA3.1 should be “pcDNA3.1” as stated everywhere

Response: Thanks, we have corrected it.

Point 9: P4 L155, the word with is repeated with with

Response: Thanks, we have corrected it.

Point 10: P4 L174, cells only positive for PI were classified as normal, should be “cells only positive for PI were classified as necrotic”.

Response: Thanks, we have corrected it.

Reviewer 2 Report

The manuscript titled "Death receptor DR5 as a proviral factor for viral entry and replication of coronavirus PEDV" described an important research on the effect of host death receptor DR5 on the entry and replication of PEDV. It was found that PEDV infection up-regulated expression of DR5 in both Vero cells and piglets. DR5 could facilitate viral replication by regulating caspase-8-dependent apoptosis. In addition, DR5 facilitates virus attachment of PEDV. The manuscript is well written and the data are well presented with in-depth discussion.

Minor revision:

1.      Fig.1A,please mark the position of DR5. In fig 1G, are the differences of PEDV titers significant at different time points with or without DR5 overexpression?

2.      Missing uncleaved caspase-3 and -8 data in Fig. 3B, and missing cleaved/uncleaved caspase-3 and uncleaved caspase-8 in Fig. 3C.

Author Response

Viruses

Manuscript ID: viruses-2056518

Title: Death receptor DR5 as a proviral factor for viral entry and replication of coronavirus PEDV

Response to Reviewer 2 Comments

Reviewer 2 comments and Suggestions for Authors

The manuscript titled "Death receptor DR5 as a proviral factor for viral entry and replication of coronavirus PEDV" described an important research on the effect of host death receptor DR5 on the entry and replication of PEDV. It was found that PEDV infection up-regulated the expression of DR5 in both Vero cells and piglets. DR5 could facilitate viral replication by regulating caspase-8-dependent apoptosis. In addition, DR5 facilitates virus attachment of PEDV. The manuscript is well written and the data are well presented with in-depth discussion.

Response: We appreciate your valuable comments.

Minor revision:

Point 1: Fig.1A,please mark the position of DR5. In fig 1G, are the differences of PEDV titers significant at different time points with or without DR5 overexpression?

Response: Thanks, The DR5 has 2 bands in the western blot, the 37 kDa band is an alternatively spliced variant of DR5, which is a common result. We added this information in our LEGENDS of Figure 1. We redrew  Fig 1G and added statistic information, PEDV titers significantly increased at different time points with or without DR5 overexpression.

Point 2: Missing uncleaved caspase-3 and -8 data in Fig. 3B, and missing cleaved/uncleaved caspase-3 and uncleaved caspase-8 in Fig. 3C.

Response: We  appreciate your constructive comments.  The deadline for paper revision is 5 days.  However, we have to wait at least 20 days for the antibodies we ordered recently due to COVID-19.  After careful consideration, we believe that the loss of this band would affect the integrity of the data, but not the conclusion of this paper.

I genuinely appreciate the great effort you've made in reviewing our manuscript. And I apologize for the situation we are in, I can't take a more active approach to deal with it now.
